# A Mixed-Method Modified Delphi Study toward Identifying Key Elements of Psychotherapy in Iran

**DOI:** 10.3390/ijerph17072514

**Published:** 2020-04-07

**Authors:** Leeba Rezaie, Shima Heydari, Ethan Paschall, Habibolah Khazaie, Dena Sadeghi Bahmani, Serge Brand

**Affiliations:** 1Sleep Disorders Research Center, Kermanshah University of Medical Sciences, Kermanshah 6719851115, Iran; rezaie.phd.ot@gmail.com (L.R.); dena.sadeghibahmani@upk.ch (D.S.B.); serge.brand@upk.ch (S.B.); 2Student Research Committee of Kermanshah University of Medical Sciences, Kermanshah 6719851115, Iran; sh.heydari88@yahoo.com; 3Clinical Psychology Doctoral Fellow, Eastern Michigan University, Ypsilanti, MI 48197, USA; epaschal@emich.edu; 4Center of Affective, Stress and Sleep Disorders, Psychiatric Clinics, University of Basel, 4001 Basel, Switzerland; 5Departments of Physical Therapy, University of Alabama at Birmingham, Birmingham, AL 35209, USA; 6Division of Sport Science and Psychosocial Health, Department of Sport, Exercise and Health, Faculty of Medicine, University of Basel, 4001 Basel, Switzerland; 7School of Medicine, Tehran University of Medical Sciences, Tehran 1416753955, Iran

**Keywords:** psychotherapy, key elements, Iran, mixed-method Delphi study

## Abstract

Purpose: In Iran, psychotherapy is regarded as an effective treatment for psychiatric disorders. However, no previous research has identified the key elements of psychotherapy that may be specific to Iranian society. The current study was conducted in an attempt to identify these elements. Methods: A mixed-method modified Delphi approach was used, taking place over several stages during 2017–2018. The first stage involved interviewing 12 experts in psychotherapy to identify key elements of psychotherapy in Iran by thematic analysis. Then, successive Delphi rounds were conducted to obtain consensus (75% agreement) from 70 psychotherapy experts on these key elements. Results: Key elements of psychotherapy were grouped into the following themes: (1) systematic education/training; (2) psychotherapist competency; (3) psychotherapy reflective of Iranian societal needs; and (4) the substrate (scientific/ethical principles) of psychotherapy. Consensus was reached during two Delphi rounds. In Delphi round 1, 52.8% of the statements reached consensus, and all remaining statements reached consensus in round 2. Conclusions: The key elements of psychotherapy in Iran are a set of conditions for the education and training of competent psychotherapists who can perform psychiatric interventions appropriate to Iranian society under supervised rules. These should serve as a framework for improving the current delivery of psychotherapy in Iran.

## 1. Introduction

Psychotherapy is an effective and evidence-based approach to the treatment of mental illness, whether used alone or in combination with pharmacotherapy. Psychotherapy is administered by a trained therapist who addresses the client’s concerns by focusing on improving the client’s understanding of his or her presenting problems and teaching new coping strategies to the client in order to reduce symptom presentation [1,2,3,4]. Psychotherapy has also been reported to lead to an enhanced sense of value for oneself, a reduction in maladaptive behaviors, and an improved overall quality of life [5].

Psychotherapy is typically employed by psychologists and psychiatrists in hospitals or outpatient clinic settings. Psychotherapy is also a key aspect of psychiatry residency training. Historically, psychotherapy began with Freud’s development of psychoanalysis at the turn of the 20th century and continued to grow as new psychological theories and short-term models of therapy were developed, such as cognitive behavioral therapy, interpersonal psychotherapy [6], and humanistic psychotherapy [3,4,7]. Today, numerous types of psychotherapy exist, including traditional approaches and third-wave behavioral approaches such as acceptance and commitment therapy (ACT) [8], mindfulness-based approaches [8,9], and internet-based psychotherapy [10,11,12,13].

The primary goal of psychotherapy is to provide a solution to a client’s presenting problems and facilitate the reduction of symptoms. However, due to the ever-changing nature of problems and the potential for differing symptom presentation over time, it is necessary that psychotherapy itself continues to adapt and change over time. In order for psychotherapy to grow and adapt as a result of an ever-growing knowledge-base and to target ever-changing problems there must be agreement among experts and practicing professionals as to the optimal direction for the future of psychotherapy. Delphi studies have been conducted every ten years since 1980 to identify the future of psychotherapy with regard to shifts in theoretical orientation and the most effective types of intervention [14,15,16]. These studies are critical in highlighting the best course of action for moving a particular field forward, preparing for the future, and forecasting potential future events in order to inform current therapeutic decisions [17,18,19,20].

In Norcross, Pfund, and Prochaska’s [21] most recent Delphi study in 2013, they examined ways in which psychotherapy will adapt and shift over the next decade and identified the areas of psychotherapy in which change would be most likely to occur by the year 2022. Additionally, Norcross and colleagues [21] reported that the primary driver of change was no longer the economy, as it had been in previous Delphi studies, but was new technology in the field of psychotherapy. These findings suggest significant changes within the field of psychotherapy within the next few years, necessitating an increased understanding of ways in which current key elements of psychotherapy are most influential with specific communities. Through an increased understanding of these key elements, the field of psychotherapy can shift its focus to aspects that are most critical for enhancement and growth, while allocating resources to elements that would benefit the most in the face of potential future changes within the field.

Iran is the second largest country in the Middle East and consists of a broad variety of ethnic groups such as Azeri, Kurdish, Gilaki, Arabic, and Balochi. Compared to countries in the same geographical area, Iran has witnessed several natural disasters and the history of eight years forced war, which appeared to have increased the prevalence of mental disorders. As described by Danaei et al. [22], the infrastructure for primary health care and education has improved dramatically within the last two decades. Psychosocial, psychological and psychiatric interventions increased to help survivors of natural disasters or veterans with post-traumatic stress disorder (PTSD); to prevent unfavorable behavior such as suicidal ideation, domestic violence, and substance abuse; or to prevent unfavorable behavior related to traffic accidents [23,24,25]. On the flip side, there are still concerns about inadequacy of mental health services in Iran. These concerns were expressed by Norton et al. [26], who mentioned in their summary of the 3rd International Anxiety Congress (Tehran, 2016) that efficacious and evidence-based mental health treatments were still limited despite the high need. Of note, transfer of international knowledge and competencies are still hampered due to political and economic barriers. Given this, issues related to the mental health infrastructure in Iran deserves particular attention.

Psychotherapy in Iran has many similarities to psychotherapy found in other countries. This is likely due to the tendency for education in Iran to follow the established patterns of education found in Western countries. To illustrate, psychiatric diagnoses are based on the DSM 5 [5]. Relatedly, Mirsalimi [27] described his experiences as an Iranian psychologist practicing in the USA and emphasized challenges due to cultural and educational similarities and differences of these two cultural environments. However, there are presently no schools that emphasize the integration of training in psychotherapy with specifically Iranian culture and social conditions. In this regard, Khodayarifard et al. [28] stated that legal regulations and registration of psychology and counseling practitioners was only initiated in 2004. Similarly, Birashk [29] mentioned that the PhD in clinical psychology was established only in 2011.

The current leading theories of psychotherapy and psychotherapeutic techniques in Iran include behavioral therapy, cognitive behavioral therapy, and psychoanalysis [30]. Despite the accessibility of medication for the treatment of psychiatric disorders, psychotherapy continues to be considered an effective treatment for patients; both psychiatrists and psychologists treat psychiatric disorders with psychotherapy, especially within outpatient clinics [31]. Typically, clients who are referred to a particular psychiatrist are treated by that psychiatrist or referred to an alternative competent psychologist. Additionally, several multidisciplinary centers exist in which psychologists and psychiatrists work together. Therefore, psychiatrists and psychologists are normally both considered to be psychotherapists in Iran. However, despite the annually increasing numbers of graduated psychologist students and psychiatrist residents, there is no formal report of the real number of psychotherapists in Iran.

Despite this increase in multidisciplinary approaches, psychotherapy in Iran continues to face significant challenges. Among the most common challenges in the country are insufficient insurance cover for psychotherapeutic intervention, poor public knowledge and awareness of psychotherapy, limited confidence in the treatment administered by psychotherapists, and a lack of organized supervision of psychotherapy, all of which lead to high rates of client drop-out—up to 80% [32,33]. If these drop-out rates are to be reduced, these issues must be addressed. However, it is insufficient merely to identify what does not work in psychotherapy, and an identification of key elements that do lead to successful treatment (and therefore ultimately those that reduce unsuccessful treatment and drop-out rates) is critical in addressing these concerns. Furthermore, we note that Asian countries have similar problems as Iran has in the concept of counseling, psychotherapy, and rehabilitation that necessitate bringing change in mental health [23,32], while there is a paucity of literature about psychotherapy challenges and system deliveries in these countries [34].

In an effort to improve our understanding and implementation of psychotherapy in Iran and Asian countries with similar health-promoting issues, first, the current study addresses aspects of its dissemination. Second, the present study could be the starting point to perform similar studies in other countries. Therefore, the study was designed to identify the key elements of effective psychotherapy within the Iranian community using the Delphi method which arrives at conclusions based on the consensus of experts in the field. In relation to this, third, the Delphi method might have the potential to describe in a more generalized fashion the framework of an evidence-based and epistemological vision [35] of how psychotherapy might be developed in Iran.

## 2. Materials and Methods

### 2.1. Study Design

The current study utilized a mixed-method modified Delphi approach to reach consensus about key elements of psychotherapy in Iran (see Figure 1). The Delphi method is an iterative a priori process in which a group of experts and stakeholders reach a structured consensus on a particular topic through a specific number of rounds that integrate controlled feedback [36,37]. The rounds involve experts in a dynamic process of data gathering and data analysis, after which they rethink and modify their opinions based on participants’ statements. In doing so, the iterative process has the potential to lead to a consensus on the specific topic [37]. Delphi studies have previously been conducted successfully to reach consensus about elements of psychotherapy in the United States [14,15,16,21]. Delphi studies are low-cost, flexible, and simple procedures used for gaining information independently and confidentially from a large number of people [38]. Additionally, Delphi studies offer unique advantages that make this a suitable approach in the current study. These advantages include anonymity, lack of sociopsychological pressure on panelists, and a higher response rate than other study techniques [17,18,19,20].

The current study was approved by the ethical committee of Kermanshah University of Medical Sciences, Kermanshah, Iran. All participants were informed about the study’s procedure and the confidentiality and the protection of their personal information.

### 2.2. Expert Identification and Semi-Structured Interviews

Purposive sampling was used to identify Iranian experts in psychotherapy. Potential experts were identified through the researchers’ own searches of information held by the Iranian psychiatric association and the Iranian Psychological Association. The experts identified satisfied the following inclusion criteria: (1) published at least five peer-reviewed papers in field of psychotherapy; (2) had a minimum of at least five years’ experience in psychotherapy; and (3) retained an established reputation in psychotherapy. Twelve experts (six psychiatrists and six psychologists) were initially contacted via email in the first stage and followed up with via telephone contact after a week if they did not respond. All 12 experts contacted by the researchers agreed to participate in the study and to work toward the development of a consensus within the field of psychotherapy.

Over a three-month period (October to December of 2017), both telephone and face-to-face semi-structured interviews were conducted with the experts to ascertain their views about key elements of psychotherapy in Iran. Of note, every year during the last week of October, the annual congress of Iranian psychiatric association is held in Tehran. Coordinating with these experts, seven face-to-face interviews were conducted during this annual congress and the remaining five were conducted via telephone. Every interview began with open-ended questions about psychotherapy in Iran and was followed by more specific questions intended to probe deeper into topics discussed during the interview. An interview guide was developed by the research team to facilitate these interviews and the content of questions being asked. The questions included on this interview guide were as follows: (1) Please tell us about your experience in psychotherapy. (2) Please give us your opinion about the current state of psychotherapy in Iran. (3) In your opinion, how can the conditions of psychotherapy be improved in Iran? (4) In your opinion, what should be the key components of psychotherapy in Iran? (5) In your opinion, how may these key components be achieved? All interviews lasted between 20–60 min and were recorded. The recorded interviews were then transcribed verbatim. Thematic analysis was conducted to extract the themes present within each interview. Following this, a survey questionnaire was then developed based on the extracted themes. This questionnaire was then sent to the experts in order to test the questionnaire’s content validity.

### 2.3. Round 1 Delphi

In April of 2018, the questionnaire was circulated to a larger group of Iranian experts in psychotherapy (*n* = 70). This larger group was composed of the 12 original experts interviewed and additional experts who had at least five years’ experience working in the field of psychotherapy. These experts were selected by snowball sampling, i.e., every originally selected expert recruited another expert. When completing the survey, the experts were asked to indicate anonymously how much they agreed with each statement. Agreement was measured on a four-point Likert scale: 1 = strongly disagree, 2 = disagree, 3 = agree, and 4 = strongly agree. Consensus was defined as experts reporting that they strongly agreed or agreed with every individual statement >75%. As consensus (>75%) was not achieved for all statements in round 1, a second Delphi round was pursued to identify any additional key elements that would help to achieve consensus.

### 2.4. Round 2 Delphi

A summary of the collated statement scores of both those that reached and that did not reach agreement was then circulated to the experts (*n* = 50). The experts were asked to repeat the survey within two weeks and either confirm their original score from Round 1 or choose a new score based on a summary of scores acquired from the other experts in the first round. Once the questionnaires were returned to the researchers, the new scores were summarized and assessed for consensus. Following this second round, all statements reached consensus (>75%). Given this, the research team discontinued the Delphi rounds.

### 2.5. Data Analysis

SPSS 22.0 (IBM Corporation, Armonk, NY, USA) for Windows was used for quantitative analyses. In both of the Delphi rounds, descriptive statistics, frequencies, percentages, and cumulative percentages were generated and used to determine the degree to which the experts agreed with each other.

### 2.6. Compliance with Ethical Standards

All authors declare that they have no conflict of interest. All study procedures were performed in accordance with the ethical standards as laid down in the 1964 Declaration of Helsinki and its later amendments. This research project was approved by the ethical committee of Kermanshah University of Medical Sciences, Kermanshah, Iran (KUMS.REC.1396.15). It is important to note that in this project a treatment intervention was not tested, and participants were not interviewed individually. Instead, psychotherapists who participated in this study were informed about the study’s procedure as well as confidentiality and the protection of their personal information. Informed consent was obtained from all individual participants included in the study.

## 3. Results

### 3.1. Semi-Structured Interview

Twelve experts (five females) participated in the semi-structured interview; five lived and worked in Tehran. The remaining experts lived and worked in various other cities in Iran. The mean number of years’ experience in the field of psychotherapy was 13.25 (*SD* = 5.57) and ranged from 6 to 25 years. All participants had at least seven publications in the field of psychotherapy. Demographic characteristics of the 12 experts are reported in Table 1.

Thematic analysis of the interviews with the experts led to the extraction of four main themes: (1) systematic education/training; (2) psychotherapist competency; (3) psychotherapy reflective of Iranian societal needs; and (4) the substrate (scientific/ethical principles) of psychotherapy. All experts reported that the education programs in psychotherapy incorporated into the curriculum of psychiatry residency and training for psychology students should include regular theoretical and practical training conducted under the supervision of skilled mentors and supervisors. Experts agreed that this training model would lead to a higher quality training for psychotherapeutic practitioners. One expert said:


*“I think the main problem is education. We need systematic education through which both psychiatry residents and clinical psychologists pass both theoretical and practical psychotherapy education. Supervision should also be emphasized by educators. Our curriculum should be revised”.*
*(p11)*

According to the second theme, experts reported that psychotherapy in Iran needs to be conducted by competent psychotherapists who have acquired scientific training and communication skills sufficient to obtain a license in psychotherapy. One participant said:


*“Unfortunately, we are confronted with psychotherapists with a lack of necessary skills to do psychotherapy. Everyone who wants to do psychotherapy should have a certification of necessary training”.*
*(p3)*

The third theme emphasized by the experts was the importance of identifying types of psychotherapeutic interventions that are based on Iranian societal needs and tailored to people from different socioeconomic classes in the country


*“We should know what our societal problems are, and what appropriate approaches exist to solve them. We need to have an assessment from the society and plan accordingly. Our culture should not be overlooked”.*
*(p8)*

The final theme of psychotherapy in Iran as reported by the experts concerned the substrate for psychotherapy. This refers to the importance of psychotherapists working in accordance with scientific and ethical principles of psychotherapy.


*“Substrate for psychotherapy in Iran is currently not optimal. A substrate for both support and supervision to psychotherapists is necessary. A powerful syndicate should exist. The principles for psychotherapists should be clarified”.*
*(p4)*

Seventy-two key element statements were subsequently extracted from the semi-structured interviews. These were then used to create the survey questionnaire.

### 3.2. Delphi Survey

Seventy participants took part in the Delphi survey; 51.4% (n = 36) were female. The average length of time spent conducting psychotherapy was 10.62 years (SD = 6.82). Of these participants, 45.7% (n = 32) reported working in Tehran and the remaining participants (54.3%; n = 38) reported working in other Iranian cities. Seventy percent (n = 49) of the participants reported holding an academic position and the remaining 30% (n = 21) reported working in private practice.

In round 1, 38 of the 72 statements (52.8%) reached consensus (>75%) among the 70 participants contacted. In round 2, 20 participants (16 psychiatrists and 4 psychologists) dropped out of the study and subsequently 50 participants (71.42% of the original sample) were contacted to take part in this round. All 50 participants contacted during round 2 completed the survey. The remaining 34 statements reached consensus in round 2, and the Delphi final list of key elements was finalized.

### 3.3. Main Areas of Agreement

#### 3.3.1. Education

The first set of key elements to emerge based on the experts’ reports related to the theme of education and understanding of how psychotherapy training should be conducted and included 22 statements. The statement that received the highest level of agreement was related to the necessity that psychiatry residents become familiar with research in psychotherapy (92.8%). The experts agreed that a psychotherapy unit should exist in each psychiatry department and that a clinical psychologist should be in attendance in each of these departments (92% agreement for both statements). Communication about psychotherapy between psychiatry departments of Iranian universities (91.9%) and weekly morning reports of psychotherapy (91%) are among the other statements that reached consensus. Table 2 shows the results of rounds 1 and 2 for education elements of psychotherapy in Iran.

#### 3.3.2. Psychotherapist Competency

Experts indicated that psychotherapist competency should be a key element of psychotherapy in Iran. Ten statements reached consensus in this area. The statement that received the highest level of agreement referred to the necessity that practitioners be members of psychotherapist associations (95.9%). Other statements that reached consensus included the requirements that practitioners of psychotherapy should be either qualified psychiatrists or clinical psychologists (80%), should have received appropriate certification for conducting psychotherapy (82%), should be aware and informed of changes in the community (84.3%), and should be aware of literature in the community regarding communication (85.7%). Agreement to a lesser extent was reached regarding the personal characteristics of psychotherapists (82.8%). Table 3 shows the results of rounds 1 and 2 for the key elements relating to the theme of competency.

#### 3.3.3. Psychotherapy Interventions

Experts agreed that psychotherapy interventions are additional key elements for psychotherapy in Iran. This theme emphasized what kind of psychotherapy is appropriate to Iranian society and how these psychotherapies should be implemented; 17 statements reflected this theme. The highest level of agreement pertained to the importance of well-controlled studies assessing the effectiveness of psychotherapy techniques (98%). Additional key elements that reached consensus included the integration of psychotherapy interventions into primary care, nursery, and primary school (90%); the importance of assessments examining the societal needs for specific interventions (81.4%); the use of short-term psychotherapy interventions (81.4%); and the use of group psychotherapy interventions (82.9%). Consensus was also achieved on the topic of increasing the number of psychotherapy sessions based on a client’s condition (77.1%). Table 4 shows the results of rounds 1 and 2 for psychotherapy interventions.

#### 3.3.4. Substrate for Psychotherapy

The experts agreed that the substrate for psychotherapy is critical theme in Iran and agreed on 23 statements in this area. The highest level of consensus was concerned the necessity for a syndicate of psychotherapy in the Ministry of Health (97.9%). Consensus was also achieved on the topics of developing and applying precise monitoring of psychotherapy education for psychiatry residents by psychiatry boards, development of guidelines for the implementation of psychotherapy that are accepted by insurance companies, and the development of guidelines for referring clients for psychotherapy (all three of these statements received 77.1% consensus). Table 5 shows the results of rounds 1 and 2 for the substrate of psychotherapy.

## 4. Discussion

This is the first mixed-method modified Delphi study to explore key elements of psychotherapy in Iran from the perspective of experts. Using semi-structured interviews with the experts, four themes relating to psychotherapy in Iran were identified: (1) systematic education/training; (2) psychotherapist competency; (3) psychotherapy reflective of Iranian societal needs; and (4) the substrate (scientific/ethical principles) of psychotherapy. A total of 72 relevant statements were surveyed in two Delphi rounds, with all 72 statements reaching consensus by the second round (52.8% in the first round and 48.3% in the second round). These findings highlight the necessity of a shift in the focus of current conditions around the practice of psychotherapy in Iran to one that acknowledges the perspectives of experts in relation to key elements of psychotherapy in the country.

Systematic education/training, with 22 relevant statements, was identified as a cluster of key elements of psychotherapy in Iran. Psychiatry residents and clinical psychology students are members of the two professions conducting most of the psychotherapy in the country. Although general education in psychotherapy is a mandatory part of the curriculum in both these fields, discrepancies exist in the level of detail present across education courses in Iran [31], highlighting the need for a consistent and structured education. The experts reached consensus on statements suggesting the need for dedicated and integrative courses throughout the country. In these statements, emphasis is placed on the quantity and quality of available education programs in psychotherapy. This would allow for both psychiatry residents and clinical psychology students to acquire the same level of necessary skills and have access to consistent training focused on providing psychotherapy. Previous research [39,40] has highlighted the importance of learning psychotherapy in psychiatry residency programs, and Kovach et al. [39] reported that, despite psychiatry residents typically reporting interest in psychotherapy, there is significant variability in the quality of training opportunities available, necessitating an examination of psychotherapy curricula. Therefore, the standardization and implementation of a consistent curriculum that improves the state of psychotherapy education within Iran is necessary for the field to progress and adequately train new professionals.

The second theme identified was psychotherapist competence, with 10 relevant statements that reached consensus. The experts agreed on the necessity for an adequate level of skills and on the personal characteristics required of psychotherapists. Using Delphi studies to explore psychotherapist competency has been reported previously [41]. Competence is defined as “the habitual and judicious use of communication, knowledge, technical skills, clinical reasoning, emotions, values, and reflection in daily practice for the benefit of the individual and community being served” [42] and is part of the competence-based movement in psychology [43,44] that dictates that a psychotherapist’s competence has a pivotal role in the outcome of treatment. In psychotherapy, the competence of a therapist has the potential to increase clients’ motivation and engagement in treatment, as well as to instill hope for a more positive outcome. Therefore, it is crucial that psychotherapists who plan to engage in psychotherapy demonstrate adequate levels of competency and undergo an assessment of their competence. It should be noted that the statements in this area concern psychotherapy in general; levels of competency specific to particular types of psychotherapy should be explored in further studies.

The theme of psychotherapy reflective of Iranian societal needs was defined the third cluster of key elements that reached consensus in this study, with 17 relevant statements. These statements included the need for identification of the types of intervention which are most appropriate to Iranian society and the best ways in which to deliver them. Additionally, this area focused on the importance of accurate assessment of societal need. Specifically, this area emphasized the importance of delivering psychotherapy interventions based on the current needs of Iranian society. Typically, these types of needs assessments in mental health programs are conducted to establish a structural base for appropriate implementation of interventions. Notably, needs assessment should be considered a dynamic and continuous process requiring repetition at regular intervals, such as the assessments conducted by Norcross et al. [21]. These results should be seen as the first step toward establishing a foundation for systematic needs assessments in psychotherapy in Iran.

The last set of key elements that reached consensus was related to the substrate (scientific/ethical principles) of psychotherapy and included 23 statements agreed upon by the experts. Inappropriate substrate has been reported as a barrier to the successful delivery of psychotherapeutic interventions [33]. The focus of this area was on the necessity to establish legal and supervisory principles for delivering psychotherapy in Iran. The area aimed to provide the substrate for psychotherapists to conduct psychotherapy in an organized manner, considering the rights of both therapist and client. This area included numerous topics which the experts agreed were vital to the appropriate delivery of psychotherapy in Iran, including the establishment of principles surrounding obtaining a license to work as a psychotherapist, guidelines for proper referral for psychotherapy, and the importance of formulating tariffs for psychotherapeutic interventions. Therefore, the results presented in the current study can be helpful in writing a protocol of requirements for an appropriate psychotherapy substrate in Iran.

### 4.1. Strengths and Limitations

As the first study to identify key elements of psychotherapy in Iran, the study has several strengths. The first is the methodology employed. Using a mixed-method modified Delphi approach provided the opportunity to use both qualitative and quantitative methods to address the research question. During the first stage, semi-structured interviews (a qualitative method) can, by their nature, help the in-depth exploration of the views expressed by an expert. During the second stage, the questionnaire survey in Delphi rounds 1 and 2 (a quantitative method), consensus within a larger group of experts can be evaluated and lead to a finalized list of key elements.

Secondly, the two most common professional groups providing psychotherapy, namely psychiatrists and clinical psychologists, were recruited from all parts of Iran, and participants in this study were from numerous locations within the country. This diversity of recruitment is important since it supports the claim that the data collected and the results describing the consensus reached can be generalized nationwide. Finally, the number of the experts (n = 70) provides a Delphi pool which can be used in further study. Despite this, dropout during the second Delphi round was observed within the current study (n = 20).

The first limitation concerns the fact that the findings concerning key elements of psychotherapy in Iran are based on Iranian experts’ views. This may be a limitation for the generalization of results to other countries, although the study was designed specifically to identify key elements of psychotherapy as they relate to Iranian culture and as reported by Iranian psychotherapy experts. However, it is possible that countries with cultural similarities can partially or fully benefit from these results. Second, the study explored key elements of psychotherapy from the experts’ points of view, while the points of view and needs of patients were not assessed. Thus, theoretically, there is the risk that experts’ statements and opinions do not match patients’ needs. Third, relatedly, the study explored the key elements of psychotherapy in Iran in a general sense; it did not address specific kinds of psychotherapeutic intervention. This statement holds particularly true, as there is not one form of psychotherapy; on the contrary, as described elsewhere more thoroughly [1,3,4], specific psychotherapeutic interventions should be chosen and applied and to the patients’ needs and psychopathology. In contrast, claims of psychotherapeutic schools that “one size fits all” do not match evidence-based and scientific research on psychotherapy anymore. Such a differentiated and client-oriented approach further takes into consideration the multifaceted variety of the Iranian population, ranging from adolescents and young adults with sleep and substance use disorders [45,46], to refugees from Afghanistan, to veterans with post-traumatic stress disorders [47], to name a few. Fourth, as mentioned, we did not perform the third round of the Delphi procedure. It is therefore possible that a third round could have yielded to an even more fine-grained pattern of results.

### 4.2. Implications and Conclusion

The current study, based on the views of experts in the field, identified key elements of psychotherapy in Iran reflecting the following four themes: systematic education, psychotherapist competency, psychotherapy reflective of Iranian societal needs, and the substrate for psychotherapy. There are several implications of the findings, the first being the pivotal role of psychotherapy education. Education was determined to be an essential component required to train competent psychotherapists, with an emphasis on the ways in which psychotherapy training could be provided adequately both for psychiatry residents and for clinical psychology students. Additionally, educators who typically work in psychotherapy departments of universities have a duty to conduct assessments of societal need for specific types of psychotherapeutic intervention; ultimately this would include assessment of the efficacy of the interventions applied. Educational elements can also play a significant role in identifying the appropriate substrate for psychotherapy, such as providing specific guidelines for psychotherapy interventions. Therefore, every program working toward an improvement in the current state of psychotherapy in Iran should place significant emphasis on the element of education, while the results of this study and the statements that reached consensus within the domain of education can be helpful in the planning and implementation of a curriculum or educational program.

Secondly, in line with the pivotal role of education, the interactions among other elements should not be overlooked. For example, the delivery of societal-needs-based psychotherapy intervention by a competent psychotherapist requires an appropriate substrate that will ensure that the therapist will continue to improve their competency. In this regard, Orlinksy and Ronnestad [48,49,50] claimed that, among others, psychotherapeutic competences in terms of the sense of professional expertise appear to increase with the increasing years of experience as a psychotherapist. Furthermore, while such experiences of life and of professional life might help to put patients’ current concerns in a broader time frame of life, there might also be the risk of biasing the patient–therapist interaction as the result of the psychotherapist’s personal issues. Additionally, competent psychotherapists who work within appropriate substrates are more likely to be aware of societal needs through an integration of their work with their education. Therefore, addressing these interactions is necessary for both psychotherapy educators and health policy makers.

Thirdly, in order fully to achieve adequate levels of every identified key element in a long-term program, continuous assessment of the field of psychotherapy within Iran is needed. In this regard, and similar to a German model [51,52], both the mental health care for the population and thoroughly evaluated and evidence-based vocational curricula for mental health care professionals would clearly improve the quality of mental health services also in Iran.

## 5. Conclusions

In conclusion, the key elements of psychotherapy in Iran are a set of conditions for psychotherapy education and training of competent psychotherapists who can perform psychiatric interventions appropriate to Iranian society under supervised rules, and expert-opinion-based research such as this study at regular intervals can help to maintain an appropriate structure for psychotherapy.

## Figures and Tables

**Figure 1 ijerph-17-02514-f001:**
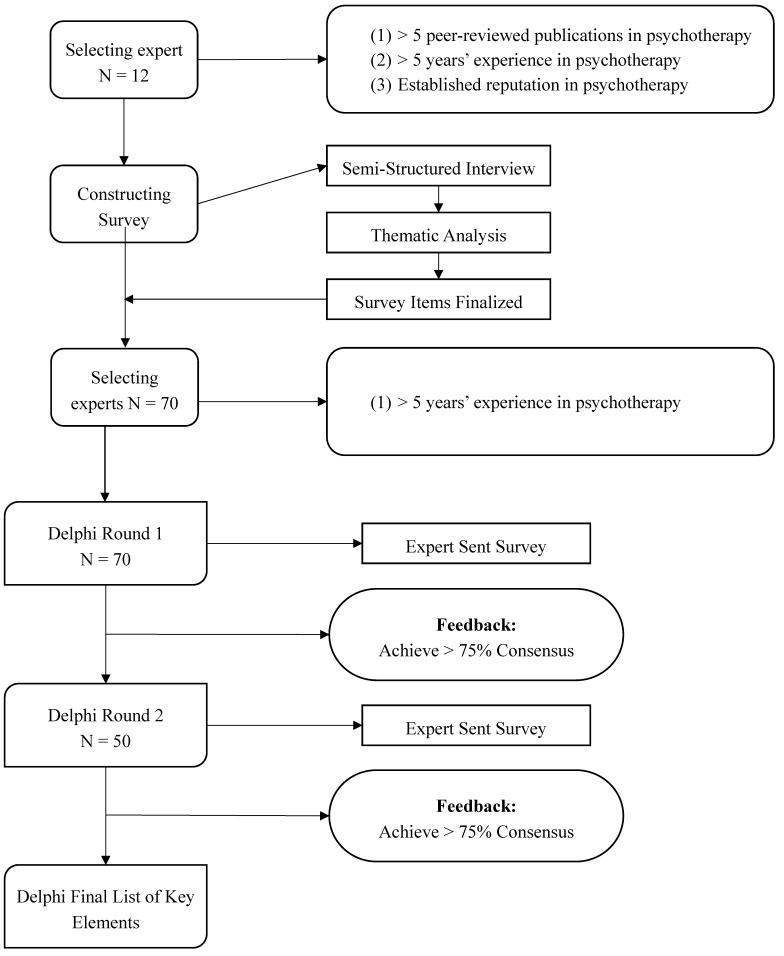
Delphi diagram of study process. This figure illustrates the steps taken in the current study.

**Table 1 ijerph-17-02514-t001:** Expert demographic characteristics from semi-structured interviews (n = 12).

Expert Number	Gender	Academic Role	Experience (Years)	Location	Number of Publications	Expert’s Area of Emphasis
1	Male	Associate professor of psychiatry	11	Tehran	10	Cognitive behavior therapy
2	Male	Assistant professor of clinical psychology	8	Tehran	7	Child and adolescence psychotherapy
3	Male	Professor of psychiatry	20	Shiraz	15	Psychoanalytical psychotherapy
4	Female	Associate professor of clinical psychology	10	Yazd	9	Child and adolescence psychotherapy
5	Female	Professor of psychiatry	25	Kerman	18	Group psychotherapy
6	Female	Professor of clinical psychology	16	Ahvaz	18	Cognitive behavior therapy
7	Male	Professor of psychiatry	15	Sanandaj	22	Transfer-based psychotherapy
8	Male	Associate professor of clinical psychology	8	Tehran	9	Cognitive behavior therapy
9	Female	Associate professor of psychiatry	6	Tabriz	10	Psychoanalytical psychotherapy
10	Male	Professor of clinical psychology	13	Tehran	12	Cognitive behavior therapy
11	Male	Associate professor of psychiatry	10	Tehran	7	Psychoanalytical psychotherapy
12	Female	Professor of clinical psychology	17	Hamadan	20	Psychoanalytical psychotherapy

**Table 2 ijerph-17-02514-t002:** Results of rounds 1 and 2 for education elements.

Education Elements	Round 1 Agreement (%)	Round 2 Agreement (%)
There should be a psychotherapy unit in each psychiatric department.	No Consensus	92%
In each psychiatric group, experienced psychotherapists should be present.	81.4%	
In each psychiatric group, at least one clinical psychologist should be present in the psychotherapy unit.	No Consensus	92%
In the psychotherapy unit, there should be a room suitable for psychotherapy.	77.1%	
In psychiatric departments, each university should communicate with other universities in Iran about psychotherapy.	84.3%	
In each psychiatric group, each university should communicate with universities in other countries about psychotherapy.	No Consensus	91.9%
Inter-university interactions and communication should lead to workshops also with adjunct professors in the field of psychotherapy.	84.3%	
The curriculum in teaching psychotherapy should change in terms of content and time.	No Consensus	88%
In each psychiatric group, a weekly morning report should be devoted to psychotherapy.	No Consensus	91%
Psychiatric residents should be familiar with some of the psychotherapy concepts from the beginning of their courses.	87.1%	
In the second year of residency, residents should attend classes devoted to psychotherapy theory.	77.2%	
During the third year, residents should practice conducting psychotherapy with a qualified professor’s supervision.	77.1%	
Psychiatric residents should learn life skills in their courses.	80%	
Psychiatric residents should experience long-term (analytical) psychotherapy in their trainings, totaling at least two cases.	No Consensus	89.8%
Psychiatry residents should learn short-term psychotherapy (such as CBT).	75.7%	
Membership of a professional body of psychotherapy should be essential for the completion of training for long-term psychotherapy.	No Consensus	86%
Psychiatric residents should be familiar with the research methods in psychotherapy.	No Consensus	92.8%
Clinical psychologists should study at universities affiliated with (and under supervision of) the Ministry of Health.	No Consensus	83.6%
Clinical psychologists should study practical psychotherapy alongside theoretical training during their education.	82.8%	
Clinical psychologists should carry out psychotherapy under supervision with a certain number of cases.	77.1%	
Psychotherapists should assess psychiatry residents and clinical psychologists during the training period and focus on individual characteristics appropriate to different types of psychotherapists.	81.4%	
Psychiatric therapists should guide psychiatry residents and clinical psychology students throughout the course of their education on the basis of individual characteristics and a variety of psychotherapists.	No Consensus	87.7%

**Table 3 ijerph-17-02514-t003:** Results of rounds 1 and 2 for psychotherapist competency.

Competency Elements	Round 1 Agreement (%)	Round 2 Agreement (%)
1. The psychotherapist should be a psychiatrist or clinical psychologist.	No Consensus	80%
2. The psychotherapist should have a certificate from a valid source (university or institution).	No Consensus	82%
3. The psychotherapist should undergo psychotherapy himself/herself.	No Consensus	87.4%
4. The psychotherapist should have the personal qualities necessary for psychotherapy (e.g., ability to communicate, empathy).	82.8%	
5. Individual psychotherapist features should be approved by several experts in psychotherapy.	No Consensus	88%
6. The psychotherapist should adhere to the principle of supervision in psychotherapy.	82.8%	
7. Psychotherapists should adhere to the principle of teamwork in psychotherapy.	77.1%	
8. The psychotherapist should be a member of a psychotherapy association.	No Consensus	95.9%
9. The psychotherapist should be aware of changes in the community.	84.3%	
10. The psychotherapist should be aware of the literature in the community regarding communication.	85.7%	

**Table 4 ijerph-17-02514-t004:** Results of rounds 1 and 2 for psychotherapy interventions.

Psychotherapy Intervention Elements	Round 1 Agreement (%)	Round 2 Agreement (%)
1. The types of psychotherapy interventions should be based on community-based needs assessments.	81.4%	
2. Psychotherapists should use short-term psychotherapy interventions.	81.4%	
3. Psychotherapists should use group psychotherapy interventions.	82.9%	
4. Psychotherapy interventions should be offered as primary care.	No Consensus	90%
5. Psychotherapy interventions should be addressed at nursery and primary schools.	No Consensus	90%
6. Short-term evidence-based psychotherapy packages should be provided.	80%	
7. The effectiveness of provided psychotherapy packages should be examined in well-controlled studies.	No Consensus	98%
8. Internet-based methods are not suitable for psychotherapy.	No Consensus	91.8%
9. Use of new methods such as mindfulness should be considered.	No Consensus	84%
10. Long-term psychotherapy is suitable for psychologically-minded people.	No Consensus	88%
11. In psychotherapy interventions, cultural sensitivities should be considered.	82.2%	
12. Psychotherapy interventions should be structured.	No Consensus	88%
13. For each psychotherapy session, reports on the content of the meeting should be provided.	77.6%	
14. Psychotherapy interventions should gradually begin for referrals.	No Consensus	88%
15. Initial psychotherapy sessions should aim solely to motivate the patient.	No Consensus	80%
16. Increase in number of psychotherapy sessions per week should be based on consideration of the client’s condition.	77.1%	
17. Psychotherapy interventions should be used in combination with medication.	83.8%	

**Table 5 ijerph-17-02514-t005:** Results of rounds 1 and 2 for substrate for psychotherapy.

Elements of Substrate for Psychotherapy	Round 1Agreement (%)	Round 2Agreement (%)
A psychotherapy branch board should exist under the psychiatric board.	No Consensus	96.4%
The board of psychiatrists should develop and apply precise, continuous monitoring criteria for psychotherapy education in psychiatry residency.	77.1%	
A syndicate for psychotherapy should exist in the Ministry of Health.	No Consensus	97.9%
Significant communication between the associations of psychology and psychiatry should exist.	82.9%	
Trade associations should have strict supervision over the license to work in psychotherapy.	82.9%	
Trade associations should require an internship in order to get a license to for work in psychotherapy.	No Consensus	85.7%
Trade associations should closely monitor the observance of principles in the work of psychotherapists.	No Consensus	89.8%
Association guidelines should require psychotherapy centers to submit monthly reports and statistics.	No Consensus	87.8%
Trade associations should have the authority to pursue legal remedies in cases of psychiatric violations.	No Consensus	89.9%
Trade unions should set out the appropriate tariffs for psychotherapy sessions.	78.6%	
Precise and fairly defined concepts of psychotherapeutic intervention should be required for trade associations to accept insurance coverage.	80%	
Trade associations should define and review the cost for psychotherapy services in the Book of Health tariffs.	82.2%	
Universities and trade associations should be committed to providing information and raising the level of information about community-based psychotherapy.	84.3%	
Universities and trade associations should be required to establish a comprehensive database of psychotherapists.	80%	
Universities and trade associations should monitor the distribution of psychotherapists nationwide.	No Consensus	94.8%
The guilds should commit to establishing branches throughout the country.	No Consensus	94.8%
Universities and trade associations should set out guidelines for psychotherapy intervention accepted by insurers.	77.1%	
Universities and guilds should be committed to developing guidelines for referring patients to psychotherapy.	77.1%	
Universities and trade associations should supervise the holding of workshops related to psychotherapy.	81.5%	
Universities and trade associations should provide access to psychotherapy services in government centers at government cost.	81.5%	
Universities and trade associations should monitor and promote the development of psychotherapists of both genders.	82.8%	
Universities and trade associations should monitor and promote the development of child psychotherapists.	84.3%	
Universities and trade associations should provide co-consulting facilities for psychotherapists.	82.9%

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
