# Peer review of "A Mixed-Method Modified Delphi Study toward Identifying Key Elements of Psychotherapy in Iran"

_ijerph, 2020, doi:10.3390/ijerph17072514_

Round 1

Reviewer 1 Report

The article is well written and structured with very good flow. The methodology used makes sense and it reflects the Depli study guidelines.

I would like to inform the authors about the following: 

It is not clear why there was a second round and what caused the experts to differentiate their answers between round 1 and round 2 in order to achieve consensus in the end. This has to be well explained because, as the text stands, this appears staged or forced.

There two major areas which need to be addressed:

First, why should readers be interested in Iran? The fact that there are gaps in psychotherapy in Iran or that there is no other study is not enough for journal publication. What is the contribution of this study to the broader scholarship which would potentially cause other researchers to read it and eventually cite it?

Second, in the abstract the authors write “However, no previous research has identified the key elements of psychotherapy that may be specific to Iranian society”. In order to address this adequately the authors need to contextualise the study better. They need to describe what it is meant by “Iranian society” in the context of the study. Is it about cultural beliefs and practices in relation to psychotherapy? Does it refer largely to the structure of the society in organisations? Are these social structures relevant to practising psychotherapy? All these have to be clarified and merged nicely together with the objectives of the study.   

Author Response

Please find the detailed point-by-point response attached as a separate file.

Reviewer 2 Report

The empirical study conducted by the authors is very interesting and particularly useful to guide the future choices of political decision-makers and of the professional associations in Iran regarding the enhancement of psychotherapy as a treatment method for the well-being of the community.

The article appears well written and clear in all sections. I have only two minor remarks for the authors:

  1. To make the usefulness of the work even more evident to Iranian and non-Iranian readers, the authors could deepen the specificities and challenges that characterize psychotherapeutic practice in Iran and also briefly address the predominant epistemological point of view embraced in reading the research problem and the study results, and, above all, the predominant epistemological vision that guided the reading of the phenomenon by experts. Here are just a few suggestions for references that could enrich the introduction and the discussion section.
  • Mirsalimi, H. (2010). Perspectives of an Iranian psychologist practicing in America. Psychotherapy: Theory, Research, Practice, Training, 47(2), 151–161. https://doi-org.ezproxy.unibg.it/10.1037/a0019754
  • Stevens, M. J., & Gielen, U. P. (2007). Toward a global psychology: Theory, research, intervention, and pedagogy. Mahwah, NJ: Lawrence Erlbaum.
    Stevens, M., & Wedding, D. (2004). The handbook of international psychology. New York: Brunner-Routledge.
  • Fourie, D. P. (1996). The research/practice gap in psychotherapy: From discovering reality to making sense. Journal of Contemporary Psychotherapy, 26, 7-22. doi: 10.1007/ BF02307702
  • Negri, A., Andreoli, G., Belotti, L., Barazzetti, A., & Martin, H. (2019). Psychotherapy trainees’ epistemological assumptions influencing research-practice integration. Research in Psychotherapy: Psychopathology, Process and Outcome, 22(3), 344-358. doi: 10.4081/ripppo.2019.397

  1. The second remark concerns the method. As far as I know, in the Delphi method second round the experts receive a anonymous feedback about all answers by all the experts involved on all items evaluated and then they are asked to re-express their evaluation on all the items proposed and not only on those that have not obtained consent in the first round. The goal is not to reach consensus at all costs on all statements and contents but to bring out consensus only on items on which there is real consensus. Having submitted only the items that did not obtain consent may have suggested that there was an implicit pressure towards consent. In fact, in the second round, all items that had not obtained consent in the first round systematically obtained it in the second round. Now, the authors could conduct a third round starting from the results of the first round and see if the results obtained are confirmed or not. Or, if this is not possible, the authors should report and comment this aspect of the method as a possible limitation of the validity of the results.

Author Response

(The authors gave the same response as above.)

Reviewer 3 Report

Metodology and methods were choosed apropriatively. A mixed method modified Delphi study was used, taking place over several stages during 2017-2018. Results: Key elements of psychotherapy were grouped into the following themes: (1) systematic education/training, (2) psychotherapist competency, (3) psychotherapy reflective of Iranian societal needs, and (4) the substrate (scientific/ethical principles) of psychotherapy.

The weaknesses of this manuscript: the conclusions were based on the consensus of experts who were psychiatrists or psychologists but not patients or clients.

The article written in an appropriate way. The results interpreted appropriately. Accept in present form.

Author Response

(The authors gave the same response as above.)

Reviewer 4 Report

Very good, interesting and highly relevant paper for establishing psychotherapy in Iran and as example for countries with lower psychotherapeutic healthcare

A few comments/questions

  1. Introduction, line 39: why Kanfer et al., a very special book on psychotherapy?
  2. fig 1: It would be helpful to indicate the number of experts per step/round
  3. page 4/line 131: “the capital city of Iran” could be deleted
  4. Some questions for discussion:
  • how many psychotherapists are there in Iran altogether? Is there any health services research data on this? Results?
  • What could psychotherapeutic care research on this look like in Iran? Perspectives? Here, the discussion on health services research could also be taken up from Germany, for example [1–3]
  • On the subject of psychotherapeutic competences, the work of Orlinsky and Ronnestad would be an interesting addition (at least for the discussion) [4–6]

Translated by www.DeepL.com/Translator (free version)

1    Literatur

1      Albani C, Blaser G, Geyer M et al. Psychotherapie und Versorgungsforschung. Psychotherapeut 2004; 49 (6): 407–414; DOI: 10.1007/s00278-004-0397-7

2      Kordy H. Versorgungsforschung. Eine wissenschaftliche und politische Herausforderung. Psychotherapeut 2008; 53: 245–253; DOI: 10.1007/s00278-008-0612-z

3      Schulz H, Barghaan D, Harfst T et al. Versorgungsforschung in der psychosozialen Medizin. Bundesgesundheitsbl - Gesundheitsforsch - Gesundheitsschutz 2006; 49 (2): 175–187; DOI: 10.1007/s00103-005-1217-0

4      Orlinsky DE, Rønnestad MH. How psychotherapists develop. A study of therapeutic work and professional growth. Washington: American Psychological Association; 2005

5      Orlinsky D, Rønnestad MH, Ambühl H et al. Psychotherapists' assessments of their development at different career levels. Psychotherapy: Theory, Research, Practice, Training 1999; 36 (3): 203–215; DOI: 10.1037/h0087772

6      Rønnestad MH, Orlinsky DE, Schröder TA et al. The professional development of counsellors and psychotherapists. Implications of empirical studies for supervision, training and practice. Couns Psychother Res 2019; 19 (3): 214–230; DOI: 10.1002/capr.12198

Author Response

(The authors gave the same response as above.)

Round 2

Reviewer 1 Report

Thank you for considering my feedback and for enhancing your paper!